# Outcomes of Liver Cancer Patients Undergoing Elective Surgery after Recovering from Mild SARS-CoV-2 Omicron Infection: A Retrospective Cohort Study

**DOI:** 10.3390/cancers15174254

**Published:** 2023-08-25

**Authors:** Yizhou Wang, Junyong Ma, Yali Wu, Shichao Zhang, Xifeng Li, Yong Xia, Zhenlin Yan, Jian Liu, Feng Shen, Xiaofeng Zhang

**Affiliations:** 1Department of Hepatic Surgery, Eastern Hepatobiliary Surgery Hospital, Navy Medical University, Shanghai 200438, China; 119337457@163.com (Y.W.); mjy_ehbh@126.com (J.M.); wuyali0505@126.com (Y.W.); zhangshichao_dr@126.com (S.Z.); lison_sh@sina.com (X.L.); xia.yong@126.com (Y.X.); zhenlinyan@sohu.com (Z.Y.); 2Graduate School, Guangdong Pharmaceutical University, Guangzhou 510006, China; 3Department of Biliary Surgery, Eastern Hepatobiliary Surgery Hospital, Navy Medical University, Shanghai 200438, China; jianliu926@163.com

**Keywords:** COVID-19, SARS-CoV-2, liver cancer, elective surgery, postoperative complications

## Abstract

**Simple Summary:**

During the early era of the COVID-19 pandemic, studies recommended delaying surgery for COVID-19 patients, given the high perioperative risk of preoperative SARS-CoV-2 infection. But in the context of widespread COVID-19 vaccination and less virulent variants, the timing of surgery for cancer patients remains unclear. In this study, we aimed to investigate the affection of preoperative mild SARS-CoV-2 Omicron infection on surgical outcomes in liver cancer patients. The average time from SARS-CoV-2 infection to surgery was 18.7 (range 7–49) days. Pre- and post-matching, there was no significant difference in preoperative characteristics and surgical outcomes between patients who had recovered from mild SARS-CoV-2 Omicron infection and those who were non infected. Postoperative major pulmonary and cardiac complications were associated with preexisting comorbidities, open surgery and COVID-19 unvaccinated, but not preoperative SARS-CoV-2 infection. Therefore, elective cancer surgery can be safely performed after recovery for patients with a history of mild SARS-CoV-2 Omicron infection.

**Abstract:**

With the emergence of new virus variants, limited data are available on the impact of SARS-CoV-2 Omicron infection on surgery outcomes in cancer patients who have been widely vaccinated. This study aimed to determine whether undergoing hepatectomy poses a higher risk of postoperative complications for liver cancer patients who have had mild Omicron infection before surgery. A propensity-matched cohort study was conducted at a tertiary liver center from 8 October 2022 to 13 January 2023. In total, 238 liver cancer patients who underwent hepatectomy were included, with 57 (23.9%) recovering from preoperative SARS-CoV-2 Omicron infection and 190 (79.8%) receiving COVID-19 vaccination. Pre- and post-matching, there was no significant difference in the occurrence of postoperative outcomes between preoperative COVID-19 recovered patients and COVID-19 negative patients. Multivariate logistic regression showed that the COVID-19 status was not associated with postoperative major pulmonary and cardiac complications. However, preexisting comorbidities (odds ratio [OR], 4.645; 95% confidence interval [CI], 1.295–16.667), laparotomy (OR, 10.572; 95% CI, 1.220–91.585), and COVID-19 unvaccinated (OR, 5.408; 95% CI, 1.489–19.633) had increased odds of major complications related to SARS-CoV-2 infection. In conclusion, liver cancer patients who have recovered from preoperative COVID-19 do not face an increased risk of postoperative complications.

## 1. Introduction

In recent years, the global outbreak of the Coronavirus Disease 2019 (COVID-19) pandemic has had a significant impact on surgical clinical practice. During the early era of the COVID-19 pandemic, given the high perioperative risk associated with SARS-CoV-2 infection, numerous studies did not recommend non-emergency surgeries for COVID-19 patients, unless the benefits of the surgery outweighed the risks of waiting [1,2,3]. The guideline also highlights the importance of individualized risk assessment, particularly for elderly, male, and comorbid patients with SARS-CoV-2 infection, who are at higher risk [2]. It was advised that their condition should be carefully assessed, and even asymptomatic COVID-19 patients should have their scheduled surgeries postponed for at least 7 weeks to minimize postoperative complications and mortality. These studies were based on the classification of COVID-19 symptoms and severity, specifically for asymptomatic patients who did not receive the vaccinations.

Amidst the COVID-19 pandemic, surgeons faced a significant challenge in the timely surgical needs from cancer patients [4]. Several international guidelines and Chinese experts have recommended delaying surgery or adapting treatment strategies for liver cancer during this pandemic [5,6,7,8]. However, most of the existing studies on the impact of COVID-19 in surgical settings are based on data from the early stages of the pandemic and may not directly apply to the current situation with SARS-CoV-2 Omicron variant, which has spread widely and in populations extensively vaccinated against COVID-19. Recent research suggests that the Omicron variant exhibits reduced virulence, with lower rates of pathogenicity, severe illness, and mortality compared to earlier strains [9]. It is also more likely to reside in the upper respiratory without causing significant lung infiltration or damage. But the specific impact of the Omicron variant on perioperative safety and surgical outcomes in vaccinated individuals remains insufficiently studied. Additionally, while previous evidence has indicated that delaying surgery may be safer during the pandemic, it is important to gather further research to determine whether surgical delays have long-term implications for cancer patients in need of elective surgery. Therefore, we hypothesized that undergoing hepatectomy posed a higher risk of postoperative complications for liver cancer patients who had recovered from mild COVID-19 during the recent SARS-CoV-2 Omicron variant pandemic in China.

## 2. Materials and Methods

### 2.1. Study Design

This single center retrospective cohort study was approved by the Institutional Ethics Board of Eastern Hepatobiliary Surgery Hospital (EHBHKY2023-02-009, 16 February 2023) and carried out in accordance with clinical research practices. The data were collected from liver cancer patients who underwent hepatectomy at our institution from 8 October 2022 to 13 January 2023. Patients who recovered from mild SARS-CoV-2 Omicron infection were matched with preoperative non-infected patients using propensity-matched analysis.

### 2.2. Participants

Patients who were diagnosed with liver cancer, including HCC and intrahepatic cholangiocarcinoma (ICC), aged 18 years or older, who underwent hepatectomy were included in the study. Follow-up was conducted up to 30 days postoperatively (the day of surgery was considered as day 0). Hepatectomy procedures were performed according to the hospital’s surgical protocols. Patients with metastatic liver cancer that did not undergo surgery of primary tumor site were included. While patients with abnormal preoperative evaluations indicating unsuitability for surgery (such as poor pulmonary function, cardiac dysfunction, etc.), those with benign tumors confirmed by postoperative pathology, those who underwent emergency surgery due to ruptured liver cancer, and patients requiring digestive tract reconstruction during surgery were excluded (Figure 1).

### 2.3. Outcome Measurement

The primary outcomes were defined as postoperative complications related to SARS-CoV-2 infection, referred to as “major complications” [10]. These included pulmonary complications (pneumonia, pulmonary embolism, and pleural effusion) and cardiac complications (myocardial injury (elevated troponin), postoperative Pre-BNP (brain natriuretic peptide), atrial fibrillation, and cardiac failure). Based on previous literature reports, these outcomes are commonly associated with COVID-19 [10,11]. The secondary outcomes included the 30-day postoperative mortality and liver-specific surgical complications following hepatectomy. These complications encompassed postoperative bleeding, liver failure, and bile leakage. Liver failure and bile leakage were defined according to the International Study Group of Liver Surgery (ISGLS) criteria. Other relevant outcomes included abdominal effusion, surgical site infection, unplanned ICU admission, nasal cannula use, electrocardiogram (ECG) monitoring, reoperation, postoperative hospital stay (counted from the day of surgery as day 0), and 30-day readmission postoperatively.

### 2.4. Diagnosis of COVID-19 Infection

From 8 October to 6 December 2022, all planned hospitalized patients underwent preoperative SARS-CoV-2 RT-PCR testing according to the local and institutional protocols. The preoperative screening protocol included the following: (1) Patients scheduled for hospitalization underwent daily nasopharyngeal swab SARS-CoV-2 testing for 2 days prior to admission. (2) Patients were surveyed for epidemiological history of contact with infected relatives within 7 days before surgery. (3) SARS-CoV-2 testing was performed on the day of admission and the following day for all patients. (4) Social distancing was encouraged for all patients before and after surgery. (5) Patients with a positive test result had their hospital admission canceled and were retested one week later. Only patients with a negative test result were rescheduled for hospitalization and surgery. (6) One caregiver per patient was allowed, and visits were restricted. (7) All healthcare staff underwent daily SARS-CoV-2 testing before starting their shifts. These liver cancer patients with preoperative negative SARS-CoV-2 tests were included as the control group for the study (COVID-19 negative). On 7 December 2022, the National Health Commission of China adjusted the epidemic policy. From 7 December 2022 to 13 January 2023, the hospital and local policies regarding admission of patients were changed, and the restriction of SARS-CoV-2 RT-PCR testing was lifted. Diagnosis of COVID-19 was based on laboratory, radiology, or clinical symptoms. Laboratory diagnosis was performed through SARS-CoV-2 RT-PCR or antigen testing from nasopharyngeal swabs, and radiological diagnosis was performed using chest CT. Clinical symptoms of mild COVID-19 primarily manifested as upper respiratory infections, including symptoms such as sore throat, cough, fever, loss of taste or smell, myalgia, and diarrhea [12]. Epidemiological investigations were conducted to determine any contact history with positive cases. Clinical diagnosis of COVID-19 was made by a senior clinician based on locally implemented protocols according to the Diagnostic and Treatment Protocol for COVID-19 (Trial Version 10) [12]. Patients suspected of having COVID-19 based on clinical or radiological methods but with negative laboratory tests were classified as non-infected. Liver cancer patients who recovered from mild COVID-19 (COVID-19 recovered) were matched with the control group, consisting of patients who underwent similar surgeries during the aforementioned time period.

### 2.5. Data Collection

Preoperative clinical parameters included age, gender, body mass index (BMI), smoking status, comorbidities (such as pulmonary, cardiovascular, hypertension, diabetes, etc.), Eastern Cooperative Oncology Group (ECOG) score, Child–Pugh classification of liver function, chest X-ray and/or CT, ECG, and COVID-19 vaccination status. Full vaccination for COVID-19 was defined as receiving two or three doses (including booster) of inactivated vaccine (CoronaVac, BBIBP-CorV, or WIBP-CorV) or recombinant protein vaccine (ZF2001). Partial vaccination was defined as receiving one dose of an inactivated vaccine or recombinant protein vaccine. The time from preoperative SARS-CoV-2 positive test to surgery was recorded. The data of American Society of Anesthesiologists (ASA) grade, operative approach, extent of resection (major hepatectomy (resection of ≥3 liver segments) vs. minor hepatectomy), tumor type, and TNM stage were also collected. Postoperative outcomes included 30-day mortality, complications, oxygen supplementation method, and readmission. Postoperative complications were classified according to the Clavien–Dindo grading system.

### 2.6. Statistical Analysis

For the comparison between preoperative COVID-19 recovered patients and negative patients, distribution tests were conducted on continuous data. Continuous data were presented as mean (SD) or median (IQR), and group differences were analyzed using *t*-tests or Wilcoxon rank-sum tests. Categorical data were presented as numbers (proportion), and analyzed using chi-square tests or Fisher’s exact tests. The preoperative, intraoperative, and 30-day postoperative outcome variables were first compared between these two groups. Then, propensity score matching was performed to match the gender, age, and ASA grade of recovered patients with negative patients in a 1:2 ratio, and the caliper value was set to 0.01. For the primary outcome of “major complications” within 30 days postoperatively, a multivariable logistic regression model was used to infer the association between the outcome and key exposure variables and covariates. Odds ratios (OR) and 95% confidence intervals (CI) were calculated to summarize the results. The model included preoperative and intraoperative factors (baseline characteristics, preexisting comorbidities, complexity of the surgery) to adjust for covariates and reduce the risk of confounding. A *p*-value < 0.05 was considered statistically significant. All statistical analyses were performed using SPSS 22.0 software (SPSS Inc., Chicago, IL, USA).

## 3. Results

### 3.1. Baseline Characteristics

As of the follow-up on 30 March 2023, a total of 238 cases were included in the cohort analysis. Among them, 57 cases were preoperative COVID-19 recovered patients and 181 cases were preoperative COVID-19 negative patients. In all, 61 cases (61/238, 25.6%) underwent chest CT examination preoperatively and 29.5% (18/61) showed pulmonary inflammatory infiltrates. The COVID-19 vaccination rate was 79.8% (190/238), with a full vaccination rate of 77.7% (185/238) (Table 1). Among the COVID-19 recovered patients, all symptomatic cases had mild symptoms (fever, pharyngitis, myalgia, etc.), which did not require hospitalization. The average time from the positive test to surgery was 18.7 ± 6.6 days, with a median time of 19.0 (range 7–49) days (Figure 2).

After matching, the CT-detected pulmonary inflammatory infiltrates in the COVID-19 recovered group was significantly higher (17.5%, 10/57) compared to the negative group (6.1%, 7/114) (*p* = 0.019). The COVID-19 vaccination rates were similar between the two groups (73.7% vs. 81.8%, *p* = 0.185) (Table 1). Apart from the differences in preoperative radiology and ECG, there were no significant differences in general preoperative and intraoperative characteristics between the matched groups.

### 3.2. Overall Outcomes

The 30-day mortality was 1.7% (4/238) in the study cohort. The incidence of major pulmonary and cardiac complications after surgery was 5.5% (13/238). Specific complications related to hepatectomy included bleeding (0.8%, 2/238), liver failure (2.1%, 5/238), and bile leakage (8%, 19/238). In the unmatched cohort, compared to the preoperative COVID-19 negative group, the preoperative COVID-19 recovered patients had a shorter hospital stay (8.3 ± 2.2 vs. 10.1 ± 5.4, *p* = 0.011), while there were no significant differences in 30-day mortality, major complications, specific complications related to hepatectomy, and other relevant outcomes (Table 2).

After matching, there were no deaths within 30 days in both groups, and modeling analysis could not be conducted for this outcome. Therefore, we focused on the occurrence of postoperative major pulmonary and cardiac complications. The incidence of pneumonia (1.8% vs. 0.9%, *p* = 0.615) and cardiac failure (0.0% vs. 0.9%, *p* = 0.478) as major complications was similar between the two groups. The incidence of specific complications related to hepatectomy was also comparable. Compared to the preoperative COVID-19 negative group, the preoperative COVID-19 recovered patients had a shorter hospital stay (8.3 ± 2.2 vs. 10.2 ± 5.9, *p* = 0.016), while there were no significant differences in other complications between the two groups (Table 2).

### 3.3. Factors Associated with Major Complications

The specific pulmonary and cardiac complications caused by SARS-CoV-2 infection were combined as a composite outcome called “major complications”. Univariate and multivariate analyses were performed on factors potentially associated with “major complications” among all patients undergoing hepatectomy (Table 3). In the univariate analysis, comorbidities, operative approach, and COVID-19 vaccination status were identified as potential influencing factors for postoperative “major complications”. However, the COVID-19 infection status was not associated with postoperative “major complications” (odds ratio [OR], 0.146; 95% confidence interval [CI], 0.013–1.654). In the multivariate analysis, pre-existing comorbidities (OR, 4.645; 95% CI, 1.295–16.667), laparotomy (OR, 10.572; 95% CI, 1.220–91.585), and non-receipt of COVID-19 vaccination (OR, 5.408; 95% CI, 1.489–19.633) were significantly associated with a higher risk of postoperative “major complications” (Figure 3).

## 4. Discussion

The occurrence of COVID-19 pandemic has had a profound impact on society, disrupting people’s lives and altering the workflow of the healthcare system. Initially, the focus was on mitigating the COVID-19 pandemic and delaying surgeries. As the variation of SARS-CoV-2 virus became more contagious but less virulent, surgical treatments gradually transitioned to prioritize surgical cases [13]. In China, the focus of pandemic prevention and control shifted from “preventing infection” to “maintaining health and preventing severe illness” [12,14]. From December 2022 to January 2023, China experienced an outbreak of the SARS-CoV-2 Omicron variant, which exhibited significant changes in transmissibility and disease severity compared to the Delta variant or the original strain [15]. Additionally, the majority of the population had received two to three doses of the COVID-19 vaccine. In this propensity-matched analysis of liver cancer patients who recovered from mild SARS-CoV-2 Omicron infection before surgery, we found no differences in postoperative 30-day mortality and major complications compared to patients without prior infection. The average time from positive SARS-CoV-2 testing to surgery was 18.7 (range 7–49) days. Importantly, COVID-19 patients with unvaccinated individuals before surgery and underlying comorbidities were associated with an increased risk of major postoperative complications.

There were no criteria on how to determine when a COVID-19 patient is no longer contagious. It was important to assess the optimal timing of surgery and appropriate preoperative evaluation for patients recovering from COVID-19. Recently, Quinn et al. found no significant correlation between postoperative outcomes of COVID-19 patients and infection within 4 weeks or 7 weeks before surgery, as well as vaccination status [16]. Another study on elective spine and gynecologic surgeries suggested that asymptomatic COVID-19 patients with incidental intraoperative SARS-CoV-2 positivity did not experience worse outcomes [17]. An updated guideline from the United Kingdom recommended that patients should avoid elective surgery within 2 weeks after SARS-CoV-2 infection [18]. Only low-risk patients who had recovered from COVID-19 and who underwent a low-risk elective surgery could proceed within 2 to 7 weeks after SARS-CoV-2 infection. Our study indicates that for liver cancer patients with good overall health and mild COVID-19 before hepatectomy, early consideration of surgery without delay may be appropriate. However, for individuals with persistent symptoms and severe or critical COVID-19, especially those with underlying comorbidities and unvaccinated individuals, the decision to proceed with elective surgery after SARS-CoV-2 infection may need to be based on the patient’s specific circumstances, weighing the risks and tumor progression associated with such delays.

Due to the different types of cancer, stage, treatment, tumor heterogeneity, and varying severity of SARS-CoV-2 infection, accurately stratifying the risk relationship between cancer and SARS-CoV-2 infection remains challenging. However, overall, cancer patients appear to be more susceptible to SARS-CoV-2 infection and death from COVID-19 [19]. Limited data are available regarding the association between SARS-CoV-2 infection and the prognosis of liver cancer. In a study on postoperative complications of liver and pancreatic cancer during the early era of the global COVID-19 pandemic conducted by the COVIDSurg Collaborative, patients with concurrent SARS-CoV-2 infection had higher postoperative mortality rates (9.4%, 12/127 vs. 2.6%, 49/1911) and complications (29.1%, 37/127 vs. 13.2%, 253/1911) compared to those without infection [20]. However, the overall mortality rate for liver cancer was 2.0% (22/1080), and in further stratified analysis, the postoperative mortality rates were comparable between the two groups (6.1%, 3/49 vs. 1.8%, 19/1031). Our study shows an overall 30-day postoperative mortality rate of 1.7% (4/238) for liver cancer patients, and there were no differences in mortality rates between the pre-matched and matched groups, which is consistent with the literature. Among the four deceased patients, two died from postoperative SARS-CoV-2 infection-related pulmonary inflammation and postoperative intra-abdominal infection combined with COVID-19, while the other two died from postoperative liver failure and postoperative bile leakage leading to intra-abdominal infection. These results highlight the need to protect elective surgery patients from SARS-CoV-2 infection during the perioperative period and promote better postoperative recovery.

Although previous observational studies have shown a considerable elevated incidence of postoperative pneumonia (7.7–21.5%) in patients infected with SARS-CoV-2 either before or peri-operatively [1,2,3,4,21,22,23]. Several recent studies showed that the incidence of postoperative complications following perioperative SARS-CoV-2 infection was not high, particularly in asymptomatic individuals [24,25,26,27]. Our study also validated the above findings, and our postoperative hospital stay was relatively shorter compared to uninfected individuals. This may be attributed to: (1) the reduced virulence of SARS-CoV-2 Omicron and widespread COVID-19 vaccination; (2) although not statistically significant, a higher proportion of infected individuals who recovered from COVID-19 were HCC patients, and they had relatively simpler surgeries with smaller resection scopes and a higher preference for minimally invasive procedures, resulting in faster postoperative recovery; (3) patients in postoperative recovery tended to discharge earlier to reduce the risk of nosocomial SARS-CoV-2 infection.

In addition, SARS-CoV-2 Omicron has increased transmissibility compared to earlier strains, but its pathogenicity is reduced, resulting in significantly lower rates of severe illness and death. The early guidelines related to COVID-19 may not be applicable to the current population that has received widespread vaccination and is infected with the Omicron variant [28]. Although the study by Quinn et al., showed that the postoperative outcomes of patients with preoperative COVID-19 were not related to vaccination status, their study was conducted prior to the Omicron variant and had a vaccination rate of only 31.4% [16]. Recent European multicenter studies on cancer patients infected with the Omicron variant after vaccination have shown significantly reduced 14-day and 28-day mortality and complication compared to the original strain and unvaccinated individuals, but these data did not include patients with liver cancer [29]. Although liver cancer has lower immunogenicity to vaccines, a multicenter study in China on HCC patients receiving inactivated COVID-19 vaccines showed that they were safe and immunogenic [30]. Furthermore, the adverse reaction and abnormal liver function occurrence rates were low in patients with HCC and liver cirrhosis who received inactivated COVID-19 vaccines after hepatectomy [31]. In China, the first dose of the vaccine was in May 2021, the second dose in June 2021, and the booster dose at the end of December 2021 [15]. Although the time interval between COVID-19 vaccination and surgery has exceeded more than 6 months, the multivariate analysis of postoperative complications in our study showed that vaccination was still a major protective factor. The overall vaccination rate in our cohort study was 79.8% (190/238), with a 77.7% (185/238) rate of full COVID-19 vaccine administration, and 56.7% (135/238) received the booster dose (Table 1). Further research is needed to determine whether booster doses improve immune response or if surgery affects immune reactivation. Additionally, since the number of individuals in the subgroups based on vaccine types is small, we will expand the population and conduct further analyses in multicenter settings to explore whether there are differences in such subgroups and their impact on postoperative complications and prognosis.

However, our current study has certain limitations. Firstly, the cohort is limited by its small sample size, retrospective nature, single center, and only including patients with liver cancer undergoing elective surgery after mild SARS-CoV-2 Omicron infection. This may restrict our ability to analyze meaningful statistical differences and perform subgroup analyses, and the results may not be applicable to patients requiring emergency surgery or those with moderate or severe COVID-19. Further research with larger populations in multicenter settings is needed in the future. Secondly, due to China’s previous “Zero COVID-19” policy implemented for nearly 3 years, some patients who had previously been infected with SARS-CoV-2 may not have received a formal diagnosis or may have concealed their medical history, resulting in their classification as COVID-19 negative. Although all surgical patients were required to provide this medical history, this study may not have captured all preoperative SARS-CoV-2 infected individuals. Thirdly, the timing of the occurrence of COVID-19-related complications was not collected, so it remains unknown whether the increase in complications in some COVID-19 negative patients after surgery was due to postoperative SARS-CoV-2 infection or vice versa. Furthermore, the analysis omitted critical data, including infection rate, morbidity, and mortality of all candidates. Additionally, specifics pertaining to dropout rates, treatment modifications, and surgical procedures among the potentially resectable candidates due to SARS-CoV-2 infection were not incorporated into the study. Finally, we focused on COVID-19-related major complications and mortality, but did not analyze the impact factors on liver surgery-specific complications such as postoperative bleeding, liver failure, bile leakage, and other adverse outcomes.

## 5. Conclusions

In conclusion, with the variation of SARS-CoV-2, future pandemics may present as localized and periodic tensions. Our study suggests that liver cancer patients who have recovered from SARS-CoV-2 Omicron infection have a similar likelihood of postoperative complications. Specifically, for patients with mild COVID-19 who have received vaccination and do not have comorbidities, elective cancer surgery can be safely performed.

## Figures and Tables

**Figure 1 cancers-15-04254-f001:**
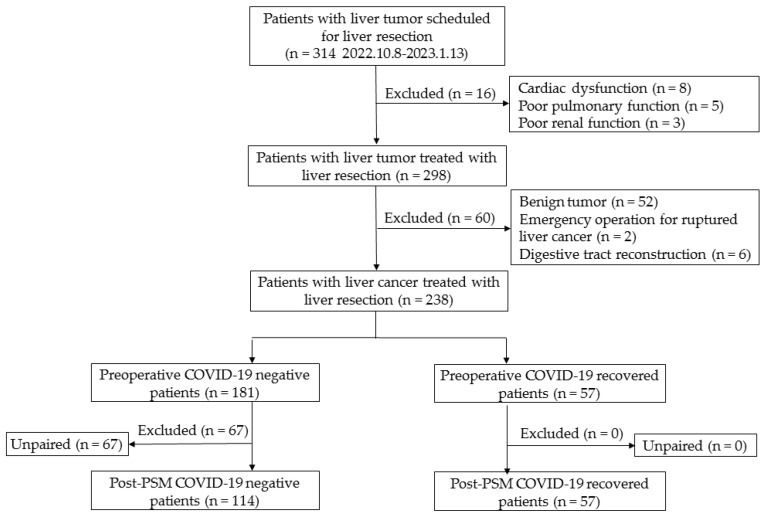
Flow chart of the study.

**Figure 2 cancers-15-04254-f002:**
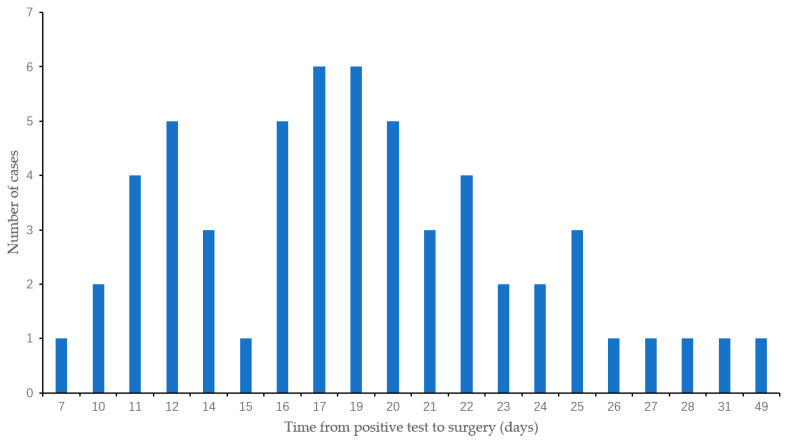
Time from SARS-CoV-2 positive test to surgery (Mean: 18.7 ± 6.6 days, Median: 19.0 (range 7–49) days).

**Figure 3 cancers-15-04254-f003:**
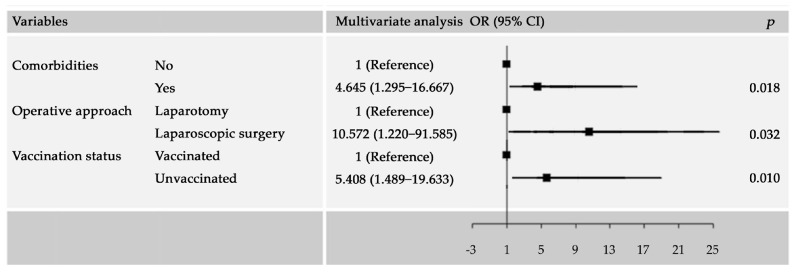
Forest plot reporting multivariable logistic regression models for major complication in patients recovered from SARS-CoV-2 Omicron infection.

**Table 1 cancers-15-04254-t001:** Descriptive statistics of eligible patients by COVID-19 status of the overall study population.

Variables	Pre-Matching			Post-Matching		
	COVID-19 Negative (*n* = 181)	COVID-19 Recovered (*n* = 57)	*p*	COVID-19 Negative (*n* = 114)	COVID-19 Recovered (*n* = 57)	*p*
Age, years (SD)	59.2 ± 11.2	56.6 ± 9.2	0.115	57.4 ± 11.8	56.6 ± 9.2	0.670
Sex			0.690			0.905
Male	135 (74.6%)	41 (71.9%)		81 (71.1%)	41 (71.9%)	
Female	46 (25.4%)	16 (28.1%)		33 (28.9%)	16 (28.1%)	
BMI, kg/m^2^ (SD)	24.4 ± 3.2	24.6 ± 3.4	0.642	24.1 ± 3.1	24.6 ± 3.4	0.332
Smoking status			0.122			0.100
Never	142 (78.5%)	50 (87.7%)		88 (77.2%)	50 (87.7%)	
Current	39 (21.5%)	7 (12.3%)		26 (22.8%)	7 (12.3%)	
Comorbidities			0.185			0.129
Pulmonary	6 (3.3%)	4 (7.0%)		2 (1.8%)	4 (7.0%)	
Cardiovascular	9 (5.0%)	0 (0.0%)		4 (3.5%)	0 (0.0%)	
Hypertension	54 (29.8%)	8 (14.0%)		27 (23.7%)	8 (14.0%)	
Diabetes	25 (13.8%)	8 (14.0%)		11 (9.6%)	8 (14.0%)	
Other	12 (6.6%)	2 (3.5%)		9 (7.9%)	2 (3.5%)	
Multiple	27 (14.9%)	6 (10.5%)		14 (12.3%)	6 (10.5%)	
ECOG performance score			0.430			0.769
0	109 (60.2%)	39 (68.4%)		76 (66.7%)	39 (68.4%)	
1	70 (38.7%)	18 (31.6%)		37 (32.5%)	18 (31.6%)	
2	2 (1.1%)	0 (0.0%)		1 (0.9%)	0 (0.0%)	
Child–Pugh class			0.386			0.156
A	180 (99.4%)	56 (98.2%)		114 (100.0%)	56 (98.2%)	
B	1 (0.6%)	1 (1.8%)		0 (0.0%)	1 (1.8%)	
Chest X-ray						
Normal	107 (59.1%)	23 (40.4%)	0.013 *	73 (64.0%)	23 (40.4%)	0.003 *
Abnormal	31 (17.1%)	13 (22.8%)	0.335	15 (13.2%)	13 (22.8%)	0.108
Not performed	43 (23.8%)	21 (36.8%)	0.052	26 (22.8%)	21 (36.8%)	0.053
Thorax CT						
Normal	3 (1.7%)	3 (5.3%)	0.130	2 (1.8%)	3 (5.3%)	0.199
Pulmonary infiltration	8 (4.4%)	10 (17.5%)	0.001 *	7 (6.1%)	10 (17.5%)	0.019 *
Other abnormal	20 (11.0%)	17 (29.8%)	<0.001 *	9 (7.9%)	17 (29.8%)	<0.001 *
Not performed	150 (82.9%)	27 (47.4%)	<0.001 *	96 (84.2%)	27 (47.4%)	<0.001 *
ECG						
Normal	80 (44.2%)	39 (68.4%)	0.001 *	50 (43.9%)	39 (68.4%)	0.002 *
ST-T change	37 (20.4%)	9 (15.8%)	0.438	24 (21.1%)	9 (15.8%)	0.411
Other abnormal	61 (33.7%)	9 (15.8%)	0.010 *	39 (34.2%)	9 (15.8%)	0.011 *
Not performed	3 (1.7%)	0 (0.0%)	0.328	1 (0.9%)	0 (0.0%)	0.478
Vaccination status			0.185			0.137
Full	145 (80.1%)	40 (70.2%)		95 (83.3%)	40 (70.2%)	
Partial	3 (1.7%)	2 (3.5%)		2 (1.8%)	2 (3.5%)	
Unvaccinated	32 (18.2%)	15 (26.3%)		17 (14.9%)	15 (26.3%)	
ASA grade			0.360			0.973
1	105 (58.0%)	39 (68.4%)		76 (66.7%)	39 (68.4%)	
2	73 (40.3%)	17 (29.8%)		36 (31.6%)	17 (29.8%)	
3	3 (1.7%)	1 (1.8%)		2 (1.8%)	1 (1.8%)	
Operative approach			0.413			0.350
Laparotomy	123 (68.0%)	42 (73.7%)		76 (66.7%)	42 (73.7%)	
Laparoscopic surgery	58 (32.0%)	15 (26.3%)		38 (33.3%)	15 (26.3%)	
Extent of resection			0.101			0.096
Minor hepatectomy	105 (58.0%)	40 (70.2%)		65 (57.0%)	40 (70.2%)	
Major hepatectomy	76 (42.0%)	17 (29.8%)		49 (43.0%)	17 (29.8%)	
Tumor type			0.270			0.221
Hepatocellular carcinoma	109 (60.2%)	41 (71.9%)		67 (58.8%)	41 (71.9%)	
Intrahepatic CC	61 (33.7%)	14 (24.6%)		43 (37.7%)	14 (24.6%)	
Other	11 (6.1%)	2 (3.5%)		4 (3.5%)	2 (3.5%)	
TNM stage			0.291			0.449
I	124 (68.5%)	32 (56.1%)		77 (67.5%)	32 (56.1%)	
II	30 (16.6%)	11 (19.3%)		19 (16.7%)	11 (19.3%)	
III	24 (13.3%)	13 (22.8%)		16 (5.3%)	13 (22.8%)	
IV	3 (1.7%)	1 (1.8%)		2 (1.8%)	1 (1.8%)	

* *p* value lower than 0.05; data presented as numbers (%), mean ± SD or median (IQR). SD, standard deviation; IQR, interquartile range; CC, cholangiocarcinoma.

**Table 2 cancers-15-04254-t002:** Postoperative outcomes of patients by COVID-19 status of the overall study population.

Variables	Pre-Matching			Post-Matching		
	COVID-19 Negative (*n* = 181)	COVID-19 Recovered (*n* = 57)	*p*	COVID-19 Negative (*n* = 181)	COVID-19 Recovered (*n* = 57)	*p*
30-day mortality	4 (2.2%)	0 (0.0%)	0.258	-	-	
Pulmonary complication						
Pneumonia	2 (1.1%)	1 (1.8%)	0.702	1 (0.9%)	1 (1.8%)	0.615
Pulmonary embolism	1 (0.6%)	0 (0.0%)		-	-	
Pleural effusion	6 (3.3%)	1 (1.8%)	0.574	4 (3.5%)	1 (1.8%)	0.521
Cardiac complication						
Elevated troponin (IQR)	0.047	0.008	0.505	0.025	0.008	0.089
	(0.001–1.9)	(0.002–0.023)		(0.001–0.143)	(0.002–0.023)	
Missing	106 (58.6%)	43 (75.4%)	0.022 *	71 (62.3%)	43 (75.4%)	0.085
Pre-BNP (IQR)	109.0 (12–409)	73.9 (21–311)	0.096	103.8 (15–354)	73.9 (21–311)	0.157
Missing	84 (46.4%)	37 (64.9%)	0.574	59 (51.8%)	37 (64.9%)	0.102
Atrial fibrillation	1 (0.6%)	0 (0.0%)	0.574	-	-	
Cardiac failure	1 (0.6%)	0 (0.0%)	0.425	1 (0.9%)	0 (0.0%)	0.478
Postoperative bleeding	2 (1.1%)	0 (0.0%)	0.205	1 (0.9%)	0 (0.0%)	0.478
Liver failure	5 (2.8%)	0 (0.0%)	0.153	3 (2.6%)	0 (0.0%)	0.217
Bile leak	17 (9.4%)	2 (3.5%)	0.765	12 (10.5%)	2 (3.5%)	0.115
Abdominal effusion	8 (4.4%)	2 (3.5%)	0.574	5 (4.4%)	2 (3.5%)	0.785
Surgical site infection	7 (3.9%)	1 (1.8%)	0.440	6 (5.3%)	1 (1.8%)	0.275
Unplanned ICU admission	2 (1.1%)	0 (0.0%)	0.425	1 (0.9%)	0 (0.0%)	0.478
Postoperative nasal cannula use	6 (3.3%)	0 (0.0%)	0.164	2 (1.8%)	0 (0.0%)	0.314
Postoperative ECG monitored	9 (5.0%)	0 (0.0%)	0.086	4 (3.5%)	0 (0.0%)	0.152
Reoperation	2 (1.1%)	0 (0.0%)	0.425	1 (0.9%)	0 (0.0%)	0.478
Hospital length of stay, days (SD)	10.1 ± 5.4	8.3 ± 2.2	0.011 *	10.2 ± 5.9	8.3 ± 2.2	0.016 *
30-day readmission	4 (2.2%)	0 (0.0%)	0.258	4 (3.5%)	0 (0.0%)	0.152

* *p* value lower than 0.05; data presented as numbers (%), mean ± SD or median (IQR). SD, standard deviation; IQR, interquartile range; BNP, brain natriuretic peptide; ICU, intensive care unit.

**Table 3 cancers-15-04254-t003:** Univariate and multivariate analysis of risk factors associated with major complication in matched groups.

Variables	Univariable OR (95% CI)	Multivariable OR (95% CI)
COVID-19		
Negative	Reference	
Recovered	0.146 (0.013–1.654)	
BMI		
<24	Reference	
≥24	0.087 (0.011–0.693)	
Comorbidities		
No	Reference	Reference
Yes	9.912 (1.613–60.928)	4.645 (1.295–16.667)
Smoking Status		
Never	Reference	
Current	2.265 (0.382–13.441)	
Chest X-ray or CT		
Normal	Reference	
Abnormal	0.000 (0.000–0.000)	
ECG		
Normal	Reference	
Abnormal	0.986 (0.194–5.011)	
Child–Pugh class		
A	Reference	
B	0.004 (0.000–1.299)	
Extent of resection		
Minor hepatectomy	Reference	
Major hepatectomy	1.698 (0.543–5.306)	
Operative approach		
Laparotomy	Reference	Reference
Laparoscopic surgery	6.408 (0.812–50.582)	10.572 (1.220–91.585)
Tumor type		
Hepatocellular carcinoma	Reference	
Other	3.924 (0.743–20.732)	
TNM stage		
I-II	Reference	
III-IV	0.825 (0.128–5.330)	
Vaccination status		
Vaccinated	Reference	Reference
Unvaccinated	46.054 (4.615–459.543)	5.408 (1.489–19.633)

Modeling the odds of major complication: pneumonia, pulmonary embolism, cardiac injury. OR, odds ratio; CI, confidence interval.

## Data Availability

The datasets used and/or analyzed during the current study are available from the corresponding author on reasonable request.

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
