# Peer review of "Outcomes of Liver Cancer Patients Undergoing Elective Surgery after Recovering from Mild SARS-CoV-2 Omicron Infection: A Retrospective Cohort Study"

_cancers, 2023, doi:10.3390/cancers15174254_

Round 1

Reviewer 1 Report

The authors present data regarding effect of COVID-19 infection on post hepatectomy morbidity and mortality in the setting of patients with HCC.  The study is a limited retrospective data set which the author's acknowledge is a limitation.  The conclusions are also not surprising in that patient's recovered from SARS COVID-19 do not appear to have an increased risk of complications as compared to those who were not infected. What would be of more value to the scientific community would be data  (at their institution) on covid infection rates in this population (HCC all comers) and morbidity/mortality as compared to HCC candidates who underwent resection as resection candidates were more likely to have better performance status. Also, if they have data regarding potential resection or locoregional therapy candidates who suffered covid prior to treatment and if there was a dropout of those patients (could not get treated) due to covid infection. Further, in resection candidates with covid, staging data regarding their disease as well as resection amount ( e.g. functional liver remnant) would also be helpful in providing context of the physiologic stress these patients underwent to better understand the impacts of covid. I believe these additional insights will better develop the story around the impacts of covid infection in this population.

Reviewer 2 Report

With their retrospective analysis the authors investigate the influence of a Covid infection on the outcome of patients who underwent liver resection.

Introduction: This part is too tedious, lines #62-73 are out of scope of the manuscript. The end of an introduction should present a hypothesis which is either proven or not by the authors.

Even for retrospective analysis is it reasonable to define a primary focus (according to the primary endpoint of the RCT), adding one or two secondary parameters. Evaluation of a colorful flower bouquet of parameters may be enticing but it is questionable from a statistical point of view. 

It is not clear if the focus of the manuscript was on the liver resection (to be expected if HCC, CCC and secondary liver tumors are summarized) or was it the Covid infection? If the Covid infection is the scope of the manuscript the authors should explain why they evaluated the HCC (with a sufficient sample size) plus CCC and secondary liver tumors, but not benign liver tumors (line #98-125).

It does not become accessible to the reader how “mild SARS-CoV-2 infection” has been defined; according to which criteria was this assessed and who did this assessment? Similar questions about the criteria “minor hepatectomy and major hepatectomy”.

Does it make sense to evaluate the hospital stay and compare it if there is no blinding and no visible criteria for discharge?

The discussion should be streamlined. The context of the paragraph between lines #312 and 333 to the title of the manuscript may be constructed only with large effort. A closer context to the focus of the manuscript would be appreciated. 

In the here presented form the manuscript is not suitable for publication in cancers

Round 2

Reviewer 1 Report

The authors have adequately addressed concerns raised during the initial review